# DiffTV: Identity-Preserved Thermal-to-Visible Face Translation via Feature Alignment and Dual-Stage Conditions

Jingyu Lin*
Monash University
Melbourne, Australia
jingyu.lin@monash.edu

Guiqin Zhao*
Kuaishou Technology
Beijing, China
zhaoguiqin20@gmail.com

Jing Xu
Monash University
Melbourne, Australia
jing.xu1@monash.edu

Guoli Wang
Tsinghua University
Beijing, China
wangguoli.2019@tsinghua.org.cn

Zejin Wang
School of Artificial Intelligence,
University of Chinese Academy of
Sciences
Beijing, China
demonszjin@gmail.com

Antitza Dantcheva
INRIA
Valbonne, France
antitza.dantcheva@inria.fr

Lan Du
Monash University
Melbourne, Australia
lan.du@monash.edu

Cunjian Chen†
Monash University
Melbourne, Australia
cunjian.chen@monash.edu

## Abstract

The thermal-to-visible (T2V) face translation task is essential for enabling face verification in low-light or dark conditions by converting thermal infrared faces into their visible counterparts. However, this task faces two primary challenges. First, the inherent differences between the modalities hinder the effective use of thermal information to guide RGB face reconstruction. Second, translated RGB faces often lack the identity details of the corresponding visible faces, such as skin color. To tackle these challenges, we introduce DiffTV, the first Latent Diffusion Model (LDM) specifically designed for T2V facial image translation with a focus on preserving identity. Our approach proposes a novel heterogeneous feature alignment strategy that bridges the modal gap and extracts both coarse- and fine-grained identity features consistent with visible images. Furthermore, a dual-stage condition injection strategy introduces control information to guide identity-preserved translation. Experimental results demonstrate the superior performance of DiffTV, particularly in scenarios where maintaining identity integrity is critical.

## CCS Concepts

• **Computing methodologies** → **Hyperspectral imaging**.

*Equal contribution.
†Corresponding author.

MM '24, October 28-November 1, 2024, Melbourne, VIC, Australia
© 2024 Copyright held by the owner/author(s).
ACM ISBN 979-8-4007-0686-8/24/10
https://doi.org/10.1145/3664647.3680635

## Keywords

Image Translation, Thermal-to-Visible, Diffusion Model, Identity-Preserving

**ACM Reference Format:**

Jingyu Lin, Guiqin Zhao, Jing Xu, Guoli Wang, Zejin Wang, Antitza Dantcheva, Lan Du, and Cunjian Chen. 2024. DiffTV: Identity-Preserved Thermal-to-Visible Face Translation via Feature Alignment and Dual-Stage Conditions. In *Proceedings of the 32nd ACM International Conference on Multimedia (MM '24), October 28-November 1, 2024, Melbourne, VIC, Australia.* ACM, New York, NY, USA, 9 pages. https://doi.org/10.1145/3664647.3680635

## 1 Introduction

Face recognition technology serves both military and commercial purposes and traditionally depends on images from the visible spectrum. Recent studies [3, 27, 33] have demonstrated very high recognition accuracies. However, these systems become less effective at night due to their dependence on sufficient lighting. Under such conditions, thermal images are advantageous as they capture the heat emitted by objects, requiring no external light source. Previous research has consistently demonstrated that thermal images maintain greater flexibility under varied lighting conditions compared to visible spectrum images [8, 41, 43]. Thus, One potential solution to this challenge could involve retraining a network to recognize faces using exclusively thermal imagery. Nevertheless, this approach is challenging because traditional CNN-based facial recognition methods [3, 33] generally require extensive datasets to perform optimally, but large-scale thermal image datasets are not widely available for public use.

In the field of image translation and face recognition, particularly under challenging lighting conditions, researchers have increasingly turned to an approach that leverages the capabilities of conditional generative models. This method begins by translating thermal images into the visible spectrum, after which traditional facial recognition algorithms are applied. GAN-based models [30, 35],

which are popular for these tasks, have shown significant potential. However, they often face complex training challenges and are vulnerable to data shortages, which can lead to issues with model stability and training convergence.

In contrast, Denoising Diffusion Probabilistic Models (DDPMs) [10] have risen to prominence in generative modeling, demonstrating superior image synthesis capabilities that surpass those of GANs, particularly in complex applications such as super-resolution and denoising [7, 38]. However, DDPMs are limited by slow inference times due to their iterative process, which can be a drawback for applications requiring rapid processing. To address this, innovations like Denoising Diffusion Implicit Models (DDIM) [28] have been developed to enhance sampling speeds. Yet, as the size of input images grows, these models still face performance bottlenecks. The integration of Latent Diffusion Models (LDM) [25] with DDIM presents an effective solution, optimizing performance for larger image sizes efficiently.

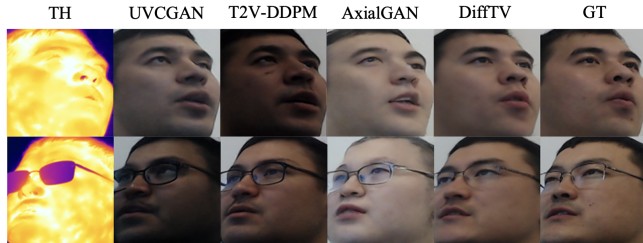

**Figure 1: Face translation outcomes of the SpeakingFaces dataset [1], TH denotes the thermal image and GT represents the Ground Truth. The compared results come from UVC-GAN [30], T2V-DDPM [22], and AxialGAN [13].**

In the context of thermal-to-visible image conversion, both GANs and DDPMs are confronted with the inherent limitations of thermal imagery. Thermal images, by their very nature, do not capture the same visual cues as visible light images, making it difficult to extract cross-modal features. Previous studies [2, 11, 15, 37] have consistently highlighted issues such as cross-modal skin color inconsistencies in existing face recognition and translation methods, which contribute to racial bias. Additionally, the subtle nuances of facial features, such as expressions and other distinctive characteristics, often become distorted or lost during the translation process. As demonstrated in Figure 1, these challenges can result in significant inconsistencies. In Figure 1, all methods except our proposed DiffTV show varying degrees of identity detail loss. This includes but is not limited to, changes in skin color, altered expressions, and facial distortions, potentially leading to final images that barely resemble the original individual, thereby undermining the goal of producing a realistic and accurate visible representation.

To address these challenges, we introduce DiffTV, the first Latent Diffusion Model (LDM) specifically designed for Thermal-to-Visible (T2V) facial image translation, which performs well in maintaining identity features. Leveraging the efficiency of operating in a lower-dimensional latent space, LDM greatly speeds up the inference process while maintaining high-quality image generation. As

illustrated in Figure 2 and Figure 3, DiffTV employs a novel heterogeneous feature alignment strategy along with a dual-stage conditional injection. This approach effectively bridges the modality gap and enhances the translation process from coarse to fine-grained details. With its superior performance, DiffTV alleviates the pressing issues of previous methods significantly, setting a new SOTA for identity-preserved facial recognition in challenging lighting conditions.

In summary, our contributions are fourfold:

- We introduce DiffTV, the first LDM-based model specifically designed for T2V facial image translation.
- We propose a novel heterogeneous feature alignment strategy that effectively extracts identity features from thermal images, ensuring the retention of identity details throughout the translation process.
- We incorporate a dual-stage conditional injection mechanism within DiffTV to facilitate a more refined identity-preserved translation from thermal images to visible images. This process allows for a granular control from coarse to fine details, significantly enhancing the model's practicality and performance in maintaining identity integrity.
- Through extensive evaluations on public datasets, we demonstrate that DiffTV outperforms existing translation methods in T2V task. This superiority is evident in scenarios demanding high fidelity in identity preservation.

## 2 Related Work

### 2.1 GAN-based Face Translation Networks

For Heterogeneous Face Recognition (HFR) tasks, GAN-based face translation networks have made significant advancements in recent years. SAGAN [4] utilizes self-attention modules to effectively synthesize visible faces from thermal images for cross-modal matching, capitalizing on discriminative information about a person's identity inherent in thermal images. Axial-GAN [13] employs axial-attention layers that harness the latest developments in transformers to model long-range dependencies, enabling the synthesis of high-resolution visible images for matching. VPGAN [20] leverages established facial priors from the visible domain to avoid learning the generation process from scratch. UVCGAN [30] improves the quality and diversity of image translations by integrating the non-local pattern learning capabilities of ViT with the cycle-consistency constraints of Cycle-Consistent GANs. HiFaceGAN [35] incrementally restores facial details using hierarchical semantic guidance, addressing the complex task of reconstructing faces with heterogeneous degradation and rich background contents. GP-UNIT [36] establishes coarse-level cross-domain correspondences with a generative prior and refines these through adversarial translations.

### 2.2 DDPM-based Face Translation Networks

To date, there are relatively few methods applying DDPM to thermal-to-visible translation, although some existing cross-modal face translation techniques could be adapted to this task. T2V-DDPM [22] is the first to present a DDPM-based solution for the Thermal-to-Visible (T2V) face translation problem. A novel inference strategy is introduced to accelerate the inference process. AT-DDPM [21] is trained using a progressive training framework and an efficient

sampling technique is introduced to reduce inference time. BBDM [17] is the first method to propose the use of Brownian Bridge diffusion processes for image-to-image translation. It directly models the translation between two domains through a bidirectional diffusion process. DiffuseIT [16] extracts intermediate keys from the ViT model and uses them as loss of content preservation. These DDPM-based approaches have demonstrated significant improvements in the quality of their output, as evidenced by metrics that assess the realism of the generated images, such as the Fréchet Inception Distance (FID) [9], and those that measure structural similarity, like the Structural Similarity Index (SSIM). In addition, there is research exploring the connection between the capacity of DDPMs to learn conditional distributions and the optimal transport theory, which seeks to find the most efficient transformation between two distributions [29].

### 2.3 Identity Preserving Image Generation

Identity-preserving image generation emphasizes the generation of images with distinct facial attributes that carry significant semantic meaning. Low-Rank Adaptation (LoRA) [12] is a widely-used, lightweight training method that involves adding a small number of additional weights into the pre-existing model to accommodate new datasets. However, LoRA's requirement for individual training for each novel character restricts its adaptability. Face0 [31] employs a technique that replaces the last three text tokens in CLIP space with the projected facial embedding, utilizing the combined embedding to guide the diffusion process. PhotoMaker [18] follows a similar strategy but enhances its capability to capture identity-specific embeddings by fine-tuning certain Transformer [5] layers within the image encoder and combining class and image embeddings. FaceStudio [34] introduces a hybrid-guidance framework for identity-preserving image synthesis, where facial embeddings are incorporated into both the visual and textual embeddings of CLIP through linear projection.

## 3 Method

In this section, we explore the methodology of our DiffTV approach for Thermal-to-Visible (T2V) image translation. Section 3.1 provides a brief summary of the thermal-to-visible task and our DiffTV, and outlines the main points and scope of the content that follows. Section 3.2 introduces the preliminaries and covers essential background information. The subsequent sections, 3.3 and 3.4, describe the Latent Diffusion Model (LDM) pipeline and the heterogeneous feature alignment strategy, respectively. Section 3.5 discusses how skin color features and Arcface identity details are combined for enhanced detail retention.

### 3.1 Overview

Assume that we have a dataset $D = \{(I_{th}^{(i)}, I_v^{(i)})_{i=1}^N\}$ where $I_{th}^{(i)}$ is the i-th thermal image and $I_v^{(i)}$ is its corresponding visible image. We aim to train a generative model $G$ parameterized by $\theta$ to minimize the distributional discrepancy between generated visible images $\hat{I}_v$ and real visible images $I_v$. Formally, our objective is to solve:

$$\min_{\theta} \mathbb{E}_{(I_{th}, I_v) \sim D} \left[ \mathcal{L} \left( I_v, G(I_{th}; \theta) \right) \right], \tag{1}$$

where $\mathcal{L}$ is a loss function measuring the fidelity of translation, so that we can ensure accurate thermal-to-visible translation. This translation process aims to bridge the modality gap between thermal and visible spectra, enabling the generation of visually informative images from thermal inputs.

In the process depicted in Figure 2, our DiffTV leverages the cutting-edge latent diffusion models tailored to convert thermal images into RGB counterparts. Alongside this core pipeline, we have developed a novel feature alignment strategy that skillfully extracts identity details from thermal images. This strategy is critical in ensuring that the generated visual outputs closely match the actual faces, thereby maintaining both consistency and accuracy. We then progressively incorporate detailed identity cues into the translation process, which is crucial for producing high-fidelity facial reconstructions.

### 3.2 Preliminaries

**Latent Diffusion**. Our method leverages the Latent Diffusion architecture [25], which efficiently executes the diffusion process with an auto-encoder [32] in a low-dimensional latent space rather than in the pixel domain. Specifically, an input image $x_i \in R^{H \times W \times 3}$ is initially transformed by the encoder into a latent form: $z_0 = \xi(x_i)$, with $z_0 \in R^{h \times w \times c}$. Here, $f = H/h = W/w$ denotes the downsampling factor and $c$ symbolizes the dimensionality of the latent space. The diffusion process adopts a denoising UNet [26] $\epsilon_\theta$ to denoise a normally-distributed noise $\epsilon$ with noisy latent $z_t$, current timestep $t$, and condition $C$. The condition $C$ is derived from the embeddings of prompts created by a specialized condition encoder. The overall training objective is defined as:

$$\mathcal{L} = \mathbb{E}_{z_t, t, C, \epsilon \sim \mathcal{N}(0,1)} \left[ ||\epsilon - \epsilon_\theta(z_t, t, C)||_2^2 \right]. \tag{2}$$

**Controllable Diffusion Models**. We utilize ControlNet [40] as an exemplar, which is capable of adding spatial control to a pre-trained diffusion model as conditions, extending beyond the capabilities of basic textual prompts. ControlNet integrates the UNet architecture from Stable Diffusion with a trainable replica of UNet. This replica features zero convolution layers within the encoder blocks and the middle block. The full process of ControlNet is executed as follows,

$$y_c = \mathcal{F}(x, \theta) + \mathcal{Z}(\mathcal{F}(x + \mathcal{Z}(c, \theta_{z1}), \theta_c), \theta_{z2}). \tag{3}$$

ControlNet sets itself apart from the foundational Stable Diffusion model through its innovative use of residuals, specifically within the $\mathcal{F}$ component, which is the UNet structure. Here, $x$ denotes the latent, $\theta$ refers to the frozen weights of the pre-trained model, and $z$ corresponds to zero convolutions, influenced by weights $\theta_{z1}$ and $\theta_{z2}$, while $\theta_c$ represents the modifiable weights within ControlNet. In essence, ControlNet encodes spatial condition information by adding residuals to UNet Block and then embeds it into the original network.

### 3.3 LDM Pipeline

Our tailored latent diffusion model is intricately composed of three fundamental components: a VQ-VAE model [32], a conditional encoder for thermal images, and a diffusion Unet model [26]. All of these constituent models are trained from scratch.

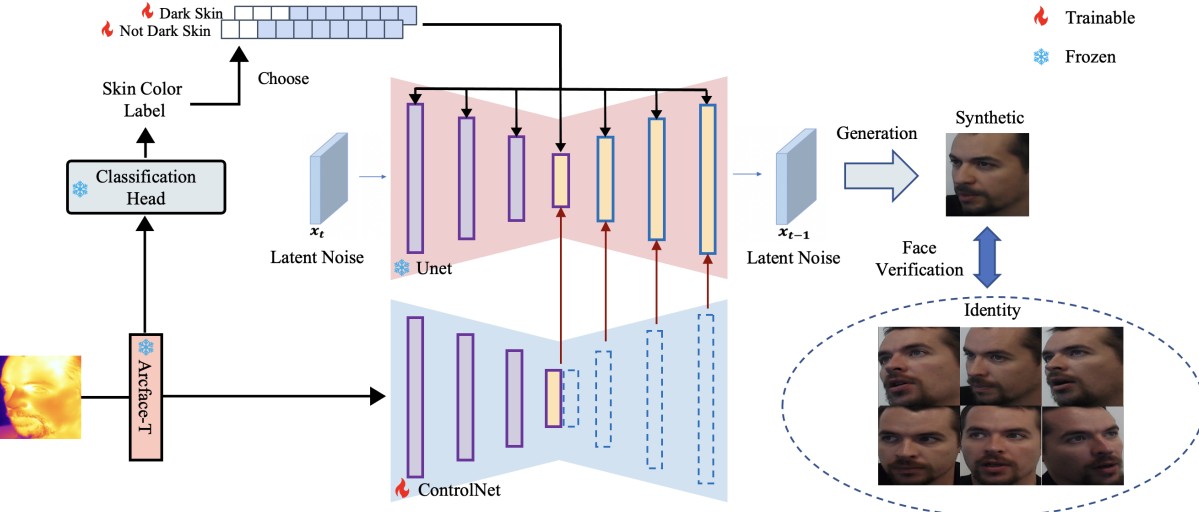

**Figure 2: Pipeline of our DiffTV. Firstly, we train a VQ-VAE model from scratch, so that the denoising process can be performed in the latent space (We note that $x_t$ is a latent noise). We employ the ArcFace model (Parameter frozen, source from heterogeneous feature alignment illustrated in Section 3.4) for fine-grained identity feature extraction from thermal images while leveraging the skin color labels to guide the generation process.**

To elaborate, we start our workflow by training a VQ-VAE model dedicated to reconstructing visible images. Following this, we proceed to train another VQ-VAE model, this time focusing on the reconstruction of thermal images. The encoder from this second VQ-VAE model is then utilized as the conditional thermal encoder, setting the stage for subsequent translation tasks.

Next, we train a DDPM UNet model within the latent space, while keeping the parameters of the initial VQ-VAE and the conditional encoder fixed. This strategy allows for the effective integration of thermal-specific features into our translation framework, significantly enhancing the fidelity and quality of the generated images.

Overall, our LDM pipeline is formally as follows:

$$\hat{I}_v = \mathcal{D}(\mathcal{E}(\epsilon_\theta(z; E_{th}(I_{th})))),\qquad(4)$$

where $\mathcal{D}, \mathcal{E}$ represents the encoder and decoder of the first-stage VQ-VAE model, $\epsilon$ and $E_{th}$ represents the diffusion noise predictor and the conditional thermal encoder.

### 3.4 Heterogeneous Feature Alignment

To effectively extract and align identity features from thermal images, we implement a heterogeneous feature alignment strategy using ArcFace, which is known for its fine-grained identity representation capabilities. ArcFace's strength lies in its attention to detailed facial features, which allows for a more nuanced understanding and encoding of identity characteristics. As shown in Figure 3, by employing this model in a trainable setup for thermal images, we take advantage of its sophisticated embedding mechanism to distill identity information from thermal data.

For the RGB images, we utilize the frozen ArcFace-S model (denoted as $ArcS$), which produces a robust embedding $e_{RGB}$ for each RGB input image $I_{RGB}$:

$$e_{RGB} = ArcS(I_{RGB}).\qquad(5)$$

On the other hand, for thermal images, we employ a trainable version of the Arcface model, noted as $ArcT$, allowing it to adapt and learn the specific features of thermal data to produce embedding $e_{Thermal}$ for each thermal input image $I_{Thermal}$:

$$e_{Thermal} = ArcT(I_{Thermal}).\qquad(6)$$

To align features from both modalities and contribute to identity preservation, we calculate the Identity Loss $L_{ID}$ as the cosine similarity between the thermal and RGB embeddings:

$$L_{ID} = 1 - \frac{e_{RGB} \cdot e_{Thermal}}{\|e_{RGB}\|\|e_{Thermal}\|}.\qquad(7)$$

The embeddings are then used to compute the skin color loss $L_{Skin}$ for skin color classification through a Classification Head (MLP), ensuring that the model discriminates between different identities effectively:

$$L_{Skin} = BCE(Class(e_{RGB}), Class(e_{Thermal})),\qquad(8)$$

where $BCE(\cdot)$ denotes for Binary CrossEntropy loss function. $Class(\cdot)$ represents the skin color classification head. By minimizing both loss functions during training, we enhance the capability of our model to maintain identity and skin-color consistency across thermal and RGB modalities.

This distilled knowledge is then applied to ensure that the generated visual representations retain the critical identity traits of the input thermal images. Through this strategy, our model manages to overcome the challenge of translating identity cues from the visual spectrum to the thermal spectrum, enhancing the fidelity and recognition accuracy of the thermal-to-visible translation.

### 3.5 Dual-stage Conditional Injection

**Skin Color Consistency**. The latent diffusion model has demonstrated remarkable superiority over previous methods. Nevertheless,

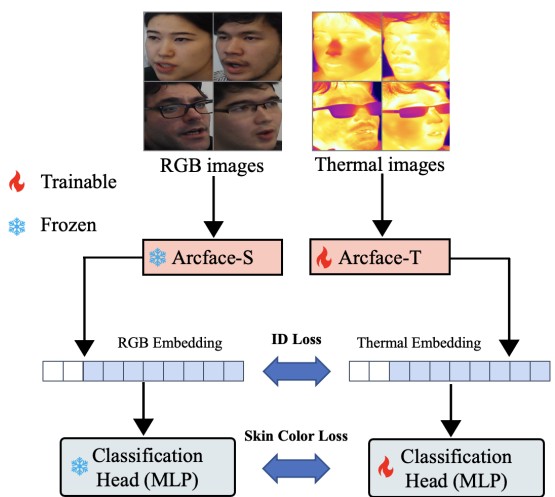

**Figure 3: Details of our proposed heterogeneous feature alignment strategy. We simultaneously aligned the Arcface embeddings and skin color classification results of both the thermal image and RGB image.**

owing to the inherent limitation that thermal infrared images do not inherently provide skin color information, some discrepancies in skin color may arise between generated faces and ground truth faces. Consequently, to deal with this challenge, we use the heterogeneous feature alignment strategy mentioned above to extract identity features directly from thermal images.

As depicted in Figure 2 and 3, the frozen Arcface uses a pre-trained backbone, capable of extracting facial embeddings. Furthermore, we categorized skin colors in the dataset into "Dark Skin" and "Not Dark Skin" and trained a classifier to achieve binary classification with almost 100 percent accuracy. We then froze the parameters of this classifier and the pre-trained model of Arcface within our Heterogeneous Feature Alignment Strategy. This approach effectively imparts the feature extraction capability of RGB faces to thermal modality faces using a concept akin to knowledge distillation.

Based on thermal faces, an almost 100 percent accuracy of binary skin color classification can be also achieved. We then initialize two learnable vectors to represent the dark- and light-skin people, respectively. Each infrared face can be assigned to a color category, so the Equation 4 is modified as follows:

$$\hat{I}_v = \mathcal{D}(\mathcal{E}(\epsilon_{\theta,p}(z; E_{th}(I_{th}), r))), \tag{9}$$

where $p$ represents the skin color encoder which is an MLP structure. $r$ denotes the skin color.

**Identity Preservertion**. Despite the alleviation of skin color inconsistency, the translated visible faces continue to exhibit a deficiency in capturing fine-grained identity details. To mitigate this issue, we introduce an additional refinement step.

Specifically, we employ ArcFace-T embedding in Section 3.4 as the identity embedding, leveraging its discriminative power to enhance the preservation of fine-grained identity features. Additionally, we integrate ControlNet [40] into our framework to fine-tune the trained model in the final stage. This refinement process ensures

a more comprehensive and accurate translation, resulting in visible faces that faithfully represent the identity details present in the thermal images.

## 4 Experiments

In this section, we provide details on experiments, datasets utilized, ablation studies performed, and comparisons with other methods.

Starting with the implementation details, we outline the experimental parameter settings in the experimental process. We will then describe the datasets used to train and test our models. Moreover, this section will include an ablation study aimed at uncovering the roles and significance of individual components within the model. By progressively simplifying the model, we can gain a clearer understanding of the contribution of each part to the final outcome. Finally, we will present the performance of our method on two datasets: SpeakingFaces [1] and ARL-VTF [24], and compare it with existing techniques.

### 4.1 Implementation Details

Our model architecture comprises a diffusion model parameterized with a base learning rate of $2.0 \times 10^{-6}$. The diffusion process is characterized by a linear noise schedule starting from 0.00085 to 0.012 over 1000 timesteps, facilitating a detailed and controlled generation process. DiffTV uses a batch size of 42 and is trained, validated, and tested on specifically curated datasets comprising thermal and visible images. The model is set to train for a maximum of 300 epochs, with validation checks every 5 epochs to monitor progress.

### 4.2 Datasets

In our experiments, we concentrate on the task of translating $128 \times 128$ thermal images to $128 \times 128$ visible images. Our primary goal is to improve the quality of the generated faces and improve the accuracy of facial recognition. Presently, there are no established benchmarks for thermal-to-visible face translation, so we reference the datasets used in the works [1, 24] for our experimentation. We conduct tests across the two distinct datasets tailored to T2V face translation. Further details on each dataset and the benchmarks used for evaluation are discussed in the following section.

**SpeakingFaces dataset [1]**: The SpeakingFaces dataset offers a vast, publicly accessible multimodal corpus suitable for machine learning studies that leverage thermal, visual, and auditory streams. It is comprised of aligned high-resolution thermal and visual spectra image streams of fully-framed faces synchronized with audio recordings of each subject speaking approximately 100 imperative phrases. Sourced from a diverse pool of 142 subjects, it presents over 13,000 synchronized samples. Notably gender-balanced and ethnically varied, subjects are recorded from multiple angles. The dataset is split into three parts: train set, validation set, and test set, with each set containing unique subjects. The synchronized samples and the wide diversity of subjects render the SpeakingFaces dataset a robust and challenging resource for research endeavors.

**ARL-VTF dataset [24]**: Unlike the SpeakingFaces dataset, the ARL-VTF dataset consists of facial images captured in the Long Wave Infrared (LWIR) modality. The dataset also provides the image capture settings for aligning the faces. However, the visible images

in ARL-VTF tend to be markedly overexposed. To address this, we employ exposure matching techniques, referencing the Speaking-Faces dataset to adjust the visible images from ARL-VTF. We create a subset of the original ARL-VTF dataset for all our experiments and choose 100 identities with different expressions as the training dataset, and data corresponding to 40 identities as the testing set. This results in a collection of 3,200 training pairs and 985 pairs for testing.

**Evaluation metrics**: For evaluating the effectiveness of our method, we utilize two different schemes [6, 20]. We prioritize evaluating the quality of the reconstructed outputs using four key metrics: Learned Perceptual Image Patch Similarity (LPIPS) [42], Fréchet Inception Distance (FID) [9], Peak Signal to Noise Ratio (PSNR) of the underlying grayscale image, and Structural Similarity Index (SSIM). Additionally, we explore face verification performance, comparing our approach against existing methods with metrics including Rank-1 accuracy and Verification Rate (VR) at False Acceptance Rates (FAR) of 1% and 0.1%. To further elucidate the performance of our DiffTV method, it is essential to understand the principles behind the evaluation metrics of VR@FAR=1% and VR@FAR=0.1%. The Verification Rate (VR) at a fixed False Acceptance Rate (FAR) is a measure used to assess the effectiveness of a biometric system, such as facial recognition. FAR represents the probability of the system incorrectly accepting a non-match as a match. When we set FAR at 1% and 0.1%, we are examining the system's ability to accurately identify individuals at these specific error rates. All facial verification experiments employ the pre-trained ArcFace facial recognition system [3].

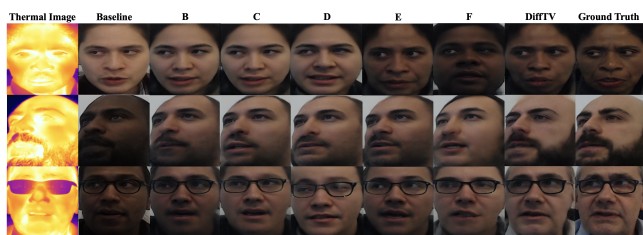

**Figure 4: Visualization results of DiffTV vs. variants. Zoom in to get a better view.**

### 4.3 Ablation Study

We assess the effectiveness of each internal module during inference and its impact on the generated results. We have set up seven sets of ablation experiments, with different variants denoted by the characters A-G, the results are shown in Table 1.

**A) Baseline**: Our baseline is an LDM conditioned on thermal images. During training, visible and thermal images are encoded into the latent space respectively via the VQVAE encoder and conditional thermal encoder, followed by the diffusion and denoising steps of DDPM. This variant has shown commendable performance across four quality metrics—achieving an FID of 35.58, LPIPS of 0.1750, PSNR of 29.19, and SSIM of 0.7082.

**B) Baseline + Skin Color Embedding**: To address the issue of skin color confusion, we introduce two trainable skin color embeddings to provide the model with a skin color prompt during

**Table 1: Ablation study on the SpeakingFaces dataset. We use A-G to denote different variants. The best result is highlighted by bold. ↑ means higher is better, and ↓ means lower is better.**

| Variants | FID↓ | LPIPS↓ | PSNR↑ | SSIM↑ |
|---|---|---|---|---|
| A | 35.58 | 0.1750 | 29.19 | 0.7082 |
| B | 34.15 | 0.1722 | 29.49 | 0.7208 |
| C | 34.33 | 0.1668 | 29.88 | 0.7206 |
| D | 33.67 | 0.1693 | 29.74 | 0.7375 |
| E | 32.16 | 0.1663 | 29.97 | 0.7629 |
| F | 34.29 | 0.1759 | 29.33 | 0.7194 |
| G (Ours) | **31.67** | **0.1553** | **30.42** | **0.7832** |

training. This embedding merges pixel-wise with latent space features through addition, applying a conditional constraint to the generation outcome. Compared to the baseline, this variant shows improvements across all metrics, validating the effectiveness of introducing skin color embedding.

**C) Baseline + ArcFace Embedding**: ArcFace, a robust facial recognition network, effectively extracts identity information from face images for classification. Our heterogeneous feature alignment strategy equips the network with the ability to focus on identity details from thermal images. Compared to the baseline, the variant enhanced with ArcFace embedding shows improvements in all metrics, even surpassing the gains of the variant in B.

**D) Baseline + ArcFace Embedding + Skin Color Embedding**: Given the effectiveness of the previous two additions, we explore a variant that combines both embeddings to constrain the model's generative results. This combination further improves model generation, with this variant's FID, LPIPS, PSNR, and SSIM reaching 33.67, 0.1693, 29.74, and 0.7375 respectively—significantly better than the baseline.

**E) Baseline + ArcFace Embedding with ControlNet**: Control-Net is known for exceptional conditional constraint capability and fine-tuning effects in generation tasks; this variant explores a better way of conditional injecting the ArcFace embedding using Control-Net. Compared to the variant in C, the results indicated marked improvements across all metrics, even exceeding the variant in D.

**F) Baseline + Skin Color Embedding with ControlNet**: Similarly, to verify whether ControlNet is also effective for skin color embedding, we explore the variant that injects the skin color embedding with ControlNet. Although the variant still shows improvements over the baseline in all metrics, it doesn't show any advantage compared to directly adding skin color embedding to LDM in variant B. Considering the strong performance of the variant in E, we conclude that ControlNet has a more potent constraint capability for fine-grained facial details, while skin color prompts are too general to showcase ControlNet's inherent strengths.

**G) DiffTV**: Integrating the previous variants, we implement a dual-stage conditional injection mechanism. By directly applying skin color information to LDM through addition, and combining it with the fine-grained identity details from ArcFace embedding by ControlNet, DiffTV achieves optimal performance. It reaches a

new SOTA with FID, LPIPS, PSNR, and SSIM scores of 31.67, 0.1553, 30.42, and 0.7832, respectively.

From the analysis above, it is evident that our model has successfully alleviated the issue of identity and skin color inconsistency prevalent in existing thermal-to-visible face translation methods. Figure 4 shows some qualitative results obtained by the variants and our DiffTV, from the figure, it is clear that the inconsistencies in identity and skin color between the original visible image and other results are significantly improved after applying DiffTV. The figure shows the effectiveness of DiffTV in dealing with identity and skin color inconsistency issues. This success is attributed to the combined feature injection methods, enhanced identity information from thermal images, and the addition of skin color priors. The utilization of each module and the methods of conditional injection have proven effective.

## 4.4 Comparison

In this section, we evaluate our method by comparing it with different generative model-based approaches for image-to-image face translation on two representative datasets.

*4.4.1 **Results on the SpeakingFaces Dataset**.* We compare our DiffTV against the prevailing thermal-to-visible methods. Such methods are predominantly categorized into two groups: GAN-based methods such as SAGAN [4] and Pix2Pix [14], and the emerging DDPM-based approaches exemplified by T2V-DDPM [22]. Our method aligns with the DDPM-based category but capitalizes on the LDM framework, renowned for its rapid inference and versatile conditional inclusion. To the best of our knowledge, DiffTV is the pioneering method of applying this framework to the thermal-to-visible task. The performance comparison presented in Table 2 clearly demonstrates DiffTV's leading edge over other methods. This is evident across a variety of metrics, including FID, which measures image diversity and quality, as well as LPIPS, PSNR, and SSIM, which are based on image similarity. The "TH" in the table represents the metrics obtained by directly comparing thermal and visible images. It's observable that GAN-based methods are generally outperformed by DDPM-based approaches. We not only employ the most advanced Diffusion Model, but also perform tailored optimizations specifically to address challenges of thermal-to-visible translation.

Given the relatively few mainstream methods specific to the thermal-to-visible task, we also compare it with recent popular methods capable of image-to-image face translation in order to validate the superiority of our proposed approach. As Table 3 shows, apart from UVCGAN [30] and CFSM [19], all listed methods are based on Diffusion Models, with BBDM [17] also utilizing the LDM framework. In particular, DiffTV significantly outperforms BBDM even when sharing the same foundational framework. Evidently, even against other Diffusion Model-based image-to-image methods, DiffTV distinctly showcases its advantages in the thermal-to-visible task. It surpasses the second best method, DiffuseIT [16], by 1.5 points in FID and 1.73 points in PSNR, while widening the gap more substantially in LPIPS and SSIM.

**Table 2: Comparison results against the prevailing methods for thermal-to-visible tasks on the SpeakingFaces dataset. The best result is highlighted by bold. ↑ means higher is better, and ↓ means lower is better.**

| Methods | FID↓ | LPIPS↓ | PSNR↑ | SSIM↑ |
|---|---|---|---|---|
| TH | 219.70 | 0.6634 | 7.88 | 0.2850 |
| Pix2Pix [14] | 70.11 | 0.5132 | 13.51 | 0.4626 |
| SAGAN [4] | 56.98 | 0.4342 | 15.11 | 0.4318 |
| GANVFS [39] | 57.19 | 0.3190 | 17.66 | 0.4905 |
| HiFaceGAN [35] | 46.77 | 0.3244 | 19.02 | 0.6233 |
| AxialGAN [13] | 43.22 | 0.2436 | 22.16 | 0.6111 |
| T2V-DDPM [22] | 39.27 | 0.2356 | 27.52 | 0.6449 |
| DiffTV(Ours) | **31.67** | **0.1553** | **30.42** | **0.7832** |

**Table 3: Comparison results against recent popular methods capable of image-to-image face translation on the SpeakingFaces dataset. The best result is highlighted by bold. ↑ means higher is better, and ↓ means lower is better.**

| Methods | FID↓ | LPIPS↓ | PSNR↑ | SSIM↑ |
|---|---|---|---|---|
| UVCGAN [30] | 52.97 | 0.2587 | 28.42 | 0.6344 |
| CFSM [19] | 49.01 | 0.2616 | 0.2798 | 0.6529 |
| Difareli [23] | 44.19 | 0.2891 | 28.11 | 0.6088 |
| BBDM [17] | 38.66 | 0.2277 | 28.83 | 0.6933 |
| DiffuseIT [16] | 33.17 | 0.2154 | 28.69 | 0.7112 |
| DiffTV(Ours) | **31.67** | **0.1553** | **30.42** | **0.7832** |

**Table 4: Comparison results on the ARL-VTF dataset. The best result is highlighted by bold. ↑ means higher is better, and ↓ means lower is better.**

| Methods | FID↓ | LPIPS↓ | PSNR↑ | SSIM↑ |
|---|---|---|---|---|
| TH | 122.70 | 0.5551 | 5.674 | 0.1095 |
| Pix2Pix [14] | 66.71 | 0.4467 | 13.12 | 0.3804 |
| SAGAN [4] | 58.22 | 0.4044 | 14.25 | 0.4490 |
| GANVFS [39] | 53.18 | 0.3924 | 13.68 | 0.4313 |
| HiFaceGAN [35] | 35.91 | 0.1937 | 19.62 | 0.6937 |
| AxialGAN [13] | 37.88 | 0.2123 | 20.04 | 0.7179 |
| T2V-DDPM [22] | 34.56 | 0.2010 | 19.70 | 0.6775 |
| DiffTV(Ours) | **30.11** | **0.1955** | **28.03** | **0.7579** |

*4.4.2 **Results on the ARL-VTF dataset**.* The ARL-VTF dataset poses a challenge for thermal-to-visible tasks because of the thermal images' blurred facial textures, making it difficult for models to extract key identity features. Nevertheless, our DiffTV method still outperforms the mainstream thermal-to-visible approaches as shown in Table 4, exhibiting superior performance. Among the comparative methods, HiFaceGAN, AxialGAN, and T2V-DDPM perform better; however, our proposed DiffTV method demonstrates a clear lead in performance metrics such as FID, PSNR, and SSIM.

TH    HiFaceGAN    AxialGAN    Difareli    DiffuseIT    T2V-DDPM    BBDM    DiffTV    GT

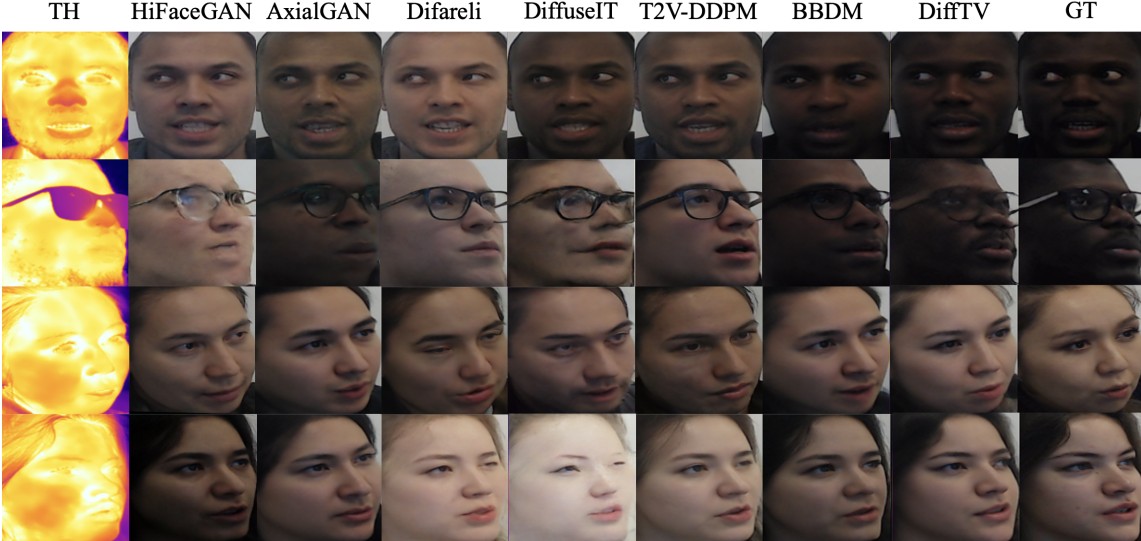

**Figure 5: Comparative visualization of face translation results. The samples source from the SpeakingFaces Dataset.**

**Table 5: Comparison results for face recognition tasks on the ARL-VTF dataset. The best result is highlighted by bold.**

| Methods | Rank-1 | VR@FAR=1% | VR@FAR=0.1% |
|---|---|---|---|
| Pix2Pix [14] | 18.88 | 5.09 | 0.33 |
| SAGAN [4] | 13.46 | 5.42 | 0.33 |
| GANVFS [39] | 21.84 | 12.97 | 2.63 |
| HiFaceGAN [35] | 65.35 | 41.33 | 20.89 |
| AxialGAN [13] | 66.67 | 42.86 | 18.62 |
| T2V-DDPM [22] | 75.37 | 43.51 | 19.87 |
| DiffTV(Ours) | **81.52** | **60.13** | **30.06** |

To provide a more quantitative evaluation of the methods, we employ two criteria, VR@FAR=1% and VR@FAR=0.1%, as previously mentioned. The quantitative outcomes are presented in Table 5. The VR@FAR metric provides a measure of the system's accuracy at these FAR levels. A higher VR@FAR value indicates a lower number of incorrectly matched identities at the given FAR, thus indicating better performance. By achieving performance gains of more than 10% in VR @ FAR = 1% and VR @ FAR = 0. 1%, our DiffTV method demonstrates a significant improvement in facial recognition accuracy from thermal images.

These experimental results not only highlight the superior performance of DiffTV in terms of quantitative metrics but also underscore the realistic nature of the generated visible images. The improvement in rank-1 accuracy by 6.15% directly reflects the method's ability to produce images that are more faithful to the original subject's appearance. This increase in accuracy is crucial for applications where the correct identification of individuals is paramount, such as security and surveillance systems.

Moreover, the substantial improvement in VR@FAR metrics indicates that the images generated by DiffTV maintain a high level of detail and accuracy, even at lower FAR thresholds where the system is less prone to false matches. This suggests that the images generated by DiffTV are not only visually convincing but also contain the necessary facial features and details required for accurate biometric analysis.

*4.4.3* ***Qualitative Analysis***. Comparative experiments underscore the efficacy of our DiffTV model, as depicted in Figure 5. Here, we present a visual comparison against other prevalent methods in the field. Not only does our approach excel in comparison to models tailored for the specific task, but it also surpasses alternative image-to-image translation techniques. From the comparison images of the generated results, it's evident that the skin color generation of most methods is not satisfactory. Among the comparative methods, only BBDM performs relatively well in this aspect. However, it still falls short in accurately restoring the details of identity. In contrast, DiffTV not only significantly improves skin color generation but also excels in preserving identity details. In addition, DiffTV uniquely facilitates the recognition of 'real' faces generated within verification systems, further demonstrating its practical superiority for applications demanding high fidelity and identity preservation.

## 5 Conclusion

In this work, we translate thermal images into visible images for face recognition in low-light conditions. We propose DiffTV, an LDM-based framework for Thermal-to-Visible (T2V) facial image translation that adeptly captures and preserves identity. DiffTV incorporates an innovative heterogeneous feature alignment strategy that effectively uses thermal imaging to extract crucial identity features. Through a dual-stage conditional approach, DiffTV ensures a detailed and identity-consistent translation from thermal to visible images. Our empirical tests across public datasets have demonstrated that DiffTV outperforms existing GAN-based and DDPM-based methods, offering a robust solution for face recognition systems operating in varying lighting conditions.

# Acknowledgments

This research is based upon work supported by the Faculty Initiatives Research, Monash University, via Contract No. 2901912, and support from the NVIDIA Academic Hardware Grant Program.

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
