# OpenReview forum: "DiffTV: Identity-Preserved Thermal-to-Visible Face Translation via Feature Alignment and Dual-Stage Conditions"
_acmmm.org/ACMMM/2024/Conference — MM2024 Poster_

### Official Review · Reviewer_gLFg · 2024-05-15

**Rating:** 5
**Confidence:** 4

**Summary:**

The article proposes a latent diffusion model (LDM) application specifically designed for thermal-to-visible (T2V) facial image conversion——DiffTV. DiffTV employs a relatively novel heterogeneous feature alignment strategy to extract key thermal imaging identity features. By using a two-stage conditional approach, it ensures detailed and identity-consistent translation from thermal images to visible images.Experiments have demonstrated that DiffTV effectively mitigates the challenges of converting thermal infrared images to visible light images for face recognition under low-light conditions.

**Strengths:**

1. The article presents an algorithm specifically designed for Thermal-to-Visible (T2V) facial image translation, which is used for facial recognition in low-light or dark conditions.Possessing a certain degree of practical value.
2. The design concept of this paper is quite ingenious, especially the heterogeneous feature alignment strategy, which conducted cross-domain transfer learning before embedding injection on self-attention, further enhancing consistency.
3. The article provides a comprehensive set of experiments and has achieved significant improvements in the accuracy of facial recognition under low-light conditions.

**Limitations:**

1. The article's narrative is somewhat vague in parts and could potentially be misleading. For instance, the article refers to the embedding processed by the MLP as "skin color loss," which is essentially a simple binary classification task with only two categories: light and dark. This is far from sufficient for defining skin color. Structurally, it appears that the elements before and after the MLP are more like coarse and fine-grained features of facial ID characteristics.
2. The article indicates that this work is the first instance of using an LDM to reconstruct high-fidelity visible facial images from thermal inputs, but it does not point out the limitations of the algorithm. The conditional injection of diffusion models + ControlNet + self-attention will inevitably lead to a decrease in efficiency, which is a rather fatal flaw for target detection and recognition. This could result in the method presented in the article being inapplicable to many practical scenarios of low-light facial recognition. If specific limitations and future research directions could be provided, it would be more meaningful for researchers and practitioners in the field.

**Suitability:**

2

---

### Official Review · Reviewer_sgUB · 2024-05-20

**Rating:** 3
**Confidence:** 3

**Summary:**

This paper introduces DiffTV, a novel Latent Diffusion Model (LDM) designed for thermal-to-visible (T2V) face translation with a focus on preserving identity. The proposed approach employs a heterogeneous feature alignment strategy and a dual-stage conditional injection mechanism to enhance the translation process. Extensive evaluations on public datasets demonstrate that DiffTV outperforms existing methods in terms of identity preservation and image quality.

**Strengths:**

1. Detailed ablation studies highlight the importance and contribution of each component of the proposed method, providing clear insights into the effectiveness of the approach.
2. The introduction of a feature alignment strategy that extracts both coarse and fine-grained identity features ensures the retention of identity details during the translation process.

**Limitations:**

1. The paper could benefit from a more detailed discussion on the inherent challenges of cross-modal translation, such as handling discrepancies between thermal and visible spectra, and how DiffTV specifically addresses these challenges.

2. How sensitive is DiffTV to the choice of hyperparameters, particularly the settings for the dual-stage conditional injection and the heterogeneous feature alignment? Have you conducted experiments to determine the optimal parameter settings?

3. How does DiffTV handle occlusions, variations in facial expressions, and different head poses in thermal images? Have you conducted experiments to evaluate its performance under such challenging conditions?

4. While the paper focuses on generative models, have you compared DiffTV’s performance with non-generative methods for thermal-to-visible translation? If so, what were the results?

**Suitability:**

2

---

### Official Review · Reviewer_dsoz · 2024-05-21

**Rating:** 3
**Confidence:** 3

**Summary:**

This paper presents DiffTV, a Latent Diffusion Model (LDM) specifically designed for Thermal-to-Visible (T2V) facial image translation. DiffTV addresses the challenges of maintaining identity features during the translation process. By operating in a lower-dimensional latent space, LDM significantly improves the efficiency of the inference process while ensuring high-quality image generation. DiffTV incorporates a novel heterogeneous feature alignment strategy and a dual-stage conditional injection mechanism, bridging the modality gap and enhancing the translation process from coarse to fine-grained details.

**Strengths:**

The use of LDM allows for efficient inference while maintaining high-quality image generation by operating in a lower-dimensional latent space.

The paper use the heterogeneous feature alignment strategy to extract identity features directly from thermal images to deal with the skin color consistency challenge and employ ArcFace-T embedding as the identity embedding, leveraging its discriminative power to enhance the preservation of fine-grained identity features.

The proposed model, DiffTV, outperforms existing methods in T2V facial image translation, particularly in challenging lighting conditions that demand high fidelity in identity preservation.

**Limitations:**

More prior work is described in Section 3, which makes the technical contribution a bit limited.

In the ablation study, variants of the experiments are described, but no pictures are given in Figure 4.

The visualization results on the ARL-VTF dataset are not shown in the paper.

Why are comparison results against the prevailing methods for thermal-to-visible face recognition tasks on the SpeakingFaces dataset not provided?

In section 4.4.1, the expression "It`s observable that GAN-based methods are generally outperformed by DDPM-based approaches." is confusing.

**Suitability:**

2

---

### Official Review · Reviewer_26Ku · 2024-05-26

**Rating:** 4
**Confidence:** 3

**Summary:**

The paper proposes a thermal-to-RGB translator based on ControlNet. To preserve the identity information in the translation process, the model utilizes the ArcFace-T extractor (the thermal image version of ArcFace) and its features as a condition to ControlNet.

**Strengths:**

- Clear performance advantage on the current SOTA

**Limitations:**

Weaknesses
- The method seems to be a slight modification of ControlNet, hence the novelty of this paper is not really significant.
- I don't see why and how the proposed model outperforms T2V-DDPM (which seems to be the current sota).
- For face identity verification, the method has been tested only on ARL-VTF. The experiment data seems not sufficient to prove the effectiveness of the proposed model. How about VIS-TH [22] dataset?
- The notations in the preliminary section on explaining ControlNet are rigorous; I could understand how ControlNet is used only after reading the original ControlNet paper. Hence the paper is not really self-contained. The notations in this subsection better need to be revised.

**Suitability:**

3

---

### Meta-Review · Area_Chair_DSmh · 2024-07-03

**Recommendation:** Accept (Poster)
**Confidence:** 5

**Metareview:**

All reviewers agree on the strengths and merits of this papers.
Some misunderstandings and objections from twop reviewers have been fully clarified by the authors in the rebuttal.
The paper has some novel insights on how to handle cross-domain human face analysis and the presented experiments well demonstrate the usefulness of the proposed approach.